# Synthesis of Microwave Functionalized, Nanostructured Polylactic Co-Glycolic Acid (*nf*PLGA) for Incorporation into Hydrophobic Dexamethasone to Enhance Dissolution

**DOI:** 10.3390/nano13050943

**Published:** 2023-03-05

**Authors:** Mohammad Saiful Islam, Somenath Mitra

**Affiliations:** Department of Chemistry and Environmental Science, New Jersey Institute of Technology, Newark, NJ 07102, USA

**Keywords:** hydrophobic drug, FDA polymer, microwave functionalization, dexamethasone, in vitro dissolution, absorption bioavailability

## Abstract

The low solubility and slow dissolution of hydrophobic drugs is a major challenge for the pharmaceutical industry. In this paper, we present the synthesis of surface-functionalized poly(lactic-co-glycolic acid) (PLGA) nanoparticles for incorporation into corticosteroid dexamethasone to improve its in vitro dissolution profile. The PLGA crystals were mixed with a strong acid mixture, and their microwave-assisted reaction led to a high degree of oxidation. The resulting nanostructured, functionalized PLGA (*nf*PLGA), was quite water-dispersible compared to the original PLGA, which was non-dispersible. SEM-EDS analysis showed 53% surface oxygen concentration in the *nf*PLGA compared to the original PLGA, which had only 25%. The *nf*PLGA was incorporated into dexamethasone (DXM) crystals via antisolvent precipitation. Based on SEM, RAMAN, XRD, TGA and DSC measurements, the *nf*PLGA-incorporated composites retained their original crystal structures and polymorphs. The solubility of DXM after *nf*PLGA incorporation (DXM–*nf*PLGA) increased from 6.21 mg/L to as high as 87.1 mg/L and formed a relatively stable suspension with a zeta potential of −44.3 mV. Octanol–water partitioning also showed a similar trend as the logP reduced from 1.96 for pure DXM to 0.24 for DXM–*nf*PLGA. In vitro dissolution testing showed 14.0 times higher aqueous dissolution of DXM–*nf*PLGA compared to pure DXM. The time for 50% (T_50_) and 80% (T_80_) of gastro medium dissolution decreased significantly for the *nf*PLGA composites; T_50_ reduced from 57.0 to 18.0 min and T_80_ reduced from unachievable to 35.0 min. Overall, the PLGA, which is an FDA-approved, bioabsorbable polymer, can be used to enhance the dissolution of hydrophobic pharmaceuticals and this can lead to higher efficacy and lower required dosage.

## 1. Introduction

Poor solubility and low bioavailability of active pharmaceutical ingredients (API) have hindered drug development, and pose many challenges for the pharmaceutical industry [1]. It is estimated that about 40% of market-approved and 90% of the development pipeline API have low aqueous solubility [2]. Such hydrophobic low-solubility APIs are classified as Biopharmaceutical Classification System (BCS) Class II and Class IV drugs, mostly weakly acidic or basic [3]. Pharmacokinetics and pharmacodynamic parameters such as drug distribution, therapeutic activity, metabolism and absorption are strongly dependent on their solubility [4]. Different approaches for solubility enhancement including particle size reduction, amorphous solid dispersions, microencapsulation, complexation, micelles, microemulsions formation, solid-state alternation, soft gel encapsulation, crystal engineering and lipid-based technologies have been used to deliver hydrophobic molecules [5,6,7,8,9]. However, they have their limitations such as alteration in the polymorph, miscibility, addition of undesirable additives and complex processing [10].

Recently, we have reported the incorporation of nano graphene oxide (nGO) and hydrophilic functionalized carbon nanotubes (fCNT) into hydrophobic APIs for enhancing dissolution in a biological medium [11,12]. The nGO or fCNTs are insoluble in water but have hydrophilic functional groups on the surface. They interact through hydrogen bonding to draw water to the drug crystals to enhance their dissolution [13,14,15]. Small amounts (1–2%) of nGO or fCNT incorporation do not alter the crystal morphology but bring about dramatic alteration in dissolution characteristics [16,17]. However, nGO and fCNTs are not FDA-approved materials and may have toxicity [18]. It will be good to have FDA-approved, biodegradable polymers that can be incorporated into hydrophobic drug crystals to enhance their dissolution.

The use of biodegradable polymers in drug delivery applications is attractive because they can break down inside the body to produce nontoxic byproducts, and the body can dispose of them [19]. Different biodegradable polymers [20,21,22] have been used for controlled [21] and sustained [23] release in therapeutic formulations as well as in different biomedical applications. In polymer-encapsulated drug delivery, the release depends on the biodegradation rate and diffusions through the polymeric matrix [24]. For example, the implants have been made with poly(lactic-co-glycolic acid) with the capability of extended drug release [25]. Dextran and alginate are other biodegradable polymers that have been widely used for sustained release [26]. Additionally, biodegradable chitosan nanoparticles have been used in thermos/pH-responsive injectable hydrogel formulations for bone and tissue engineering [27]. Moreover, different polymers have been used as excipients, surfactants, and as ingredients that facilitate processability, stability or therapeutic response [28,29,30]. Recently, many random and block polymers of polylactic acid (PLA), polyglycolic acid (PGA), polyethylene glycol (PEG), poly (vinyl alcohol) (PVA) and polycaprolactone (PCL) as well as synthetic copolymers such as polyanhydrides (PA), poly ortho esters (POE), polytetrafluorethylene (PTFE) and poly (methyl methacrylate) (PMMA) have been used for systemic drug delivery, ophthalmic formulations, and surgical implant synthesis [31,32,33]. Since the APIs are connected to a polymer and then released into a biological medium [34,35], surface functionalization plays a key role in drug loading, conjugation, immobilization, and incorporation [36].

Poly(lactic-co-glycolic acid) or PLGA, a copolymer of PLA and PGA, has found many biomedical applications because it is biodegradable, biocompatible, and bioavailable [37]. The US Food and Drug Administration (FDA)-approved PLGA nanoparticles have found successful applications for different types of parenteral, ocular, injectables and oral drug delivery [38,39]. Most of the application of PLGA has been to reduce the rate of delivery of water-soluble compounds including insulin, neurologics, vaccine, hormones, DNA, protein, and steroids [40,41]. PLGA is also used in wound cream, ointment, multivitamin and biomedical devices where drug delivery is an integral part of the process [42,43,44].

Polymer nanoparticles can be surface-functionalized to provide a high surface-to-volume ratio, allowing for maximum drug binding [45]. For example, the carboxy-terminated poly(d,l-lactide-co-glycolide)-block-poly(ethylene glycol) or PLGA-b-PEG-COOH polymer has been used in targeted drug delivery, resulting in a favorable biodistribution [46]. Research has demonstrated that functionalizing PLGA with epidermal growth factor and loading it with 5-Fluorouracil and perfluorocarbon leads to improved therapeutic outcomes by inhibiting tumor growth [47]. Furthermore, PLGA’s enhanced biodegradability makes it advantageous not only for drug dissolution but also for its efficient removal from the body without any toxicity [48]. Nanostructured titania/PLGA composites have been developed for in-tissue engineering and bone fractures, which have shown controlled biodegradation to lactic acid and/or glycolic acid through non-enzymatic hydrolysis of ester bonds [49].

Dexamethasone is a highly potent glucocorticosteroidal drug with numerous medical applications [50]. It is very hydrophobic and was widely used to treat patients during the COVID-19 pandemic for reducing adverse effects and mortality [51]. Medical research has also shown that perineural dexamethasone improves postoperative analgesia [52,53]. In the pharmaceutical field, there is significant interest in developing nanoparticle formulations of dexamethasone to enhance its efficacy, and reduce pharmacokinetic cytotoxicity. Research has shown that an anti-inflammatory dexamethasone encapsulated into a biological cell-coated membrane has enhanced therapeutic efficacy with extended in vivo delivery [54]. Liposome-encapsulated dexamethasone is another such formulation with promising results [55]. However, the major limiting factor for hydrophobic dexamethasone is its intrinsic poor solubility and dissolution properties. Developing the right nanoparticle and functionality to enhance drug solubility will greatly aid in the effectiveness of treatment using dexamethasone.

It is anticipated that PLGA, which is not water-soluble by itself, can be surface-functionalized to have a hydrophilic surface. Nanoparticles of this functionalized form can potentially be incorporated onto the surface of a hydrophobic drug crystal, which then can be conduits for bringing water in contact with the drug crystal for faster dissolution. The objective of this research was to develop water-insoluble, surface-hydrophilized, nanostructured PLGA (referred to as *nf*PLGA) that can be incorporated into API crystals to synthesize drug-*nf*PLGA composites with enhanced dissolution properties. Another objective was to carry out microwave synthesis of *nf*PLGA, which is a fast and eco-friendly process. Corticosteroid dexamethasone (DXM), which is a highly hydrophobic BCS-IV drug with low water solubility (0.089 mg/mL), was used to form the soluble composites.

## 2. Materials and Methods

### 2.1. Materials

Dexamethasone (9-fluoropregna-1,4-diene, C_22_H_29_FO_5_, 392.464 g/mol) is a synthetic anti-inflammatory corticosteroid that was previously also used as a booster medicine for COVID-19 treatment [56,57,58,59,60]. Dexamethasone was purchased from Sigma Aldrich lot # LRAC2894 (CAS # 50-02-2). poly(lactic-co-glycolic acid) or PLGA (50:50) was bought from Polysciences Inc (Warrington, PA, USA). Acetone was bought from Sigma Aldrich. Sulfuric acid, nitric acid, hydrochloric acid, and acetone were purchased from Fisher Scientific (Thermo Fisher Scientific Inc., Waltham, MA, USA). 1-Octanol was purchased from Sigma Aldrich (St. Louis, MO, USA). Phosphate buffer solution containing potassium phosphate monobasic sodium buffer solution (pH 6.0), Potassium carbonate potassium borate potassium hydroxide buffer solution (pH 10.0) were purchased from Fisher Scientific, USA. The water used in the experiment was purified with the Milli-Q plus system.

### 2.2. Synthesis of nfPLGA

The acid oxidation of PLGA was carried out in a multimode CEM microwave reactor (MARS-5, Matthews, NC, USA). The PLGA was then mixed with a 3:1 H_2_SO_4_ and HNO_3_ mixture. Next, the acid mixed poly(lactic-co-glycolic acid) or PLGA samples were placed into the microwave chamber and reacted at an applied power of 800 W (maximum 1600 W ± 15%) and frequency of 2450 MHz (wavelength λ of 0.1223642685714 m) at 60 °C with the use of IEC method. After 1.0 h of microwave induced reaction, the samples were vacuum filtered and washed to obtain the functionalized particles. After drying the *f*PLGA particles were mixed in Milli-Q water for dispersion and sonicated using high-power (110 V, 20 kHz) probe sonicator (Ultrasonic processor FS-900 N) for one hour to produce nano *f*PLGA or *nf*PLGA.

### 2.3. Synthesis of DXM–nfPLGA

An antisolvent precipitation technique was used to synthesize the DXM–*nf*PLGA composites. This was a modification of a process described before [61]. Acetone solvent was used to dissolve dexamethasone (DXM) drug. A clear solution of *nf*PLGA was also made in acetone. This was added to the dexamethasone solution dropwise and sonicated for 10.0 min. Next, the drug composite solution was placed into a cold ice bath and the antisolvent Milli-Q water was added dropwise. A white and milky suspension of the drug-polymer composite was observed during the precipitation process. The precipitate was then filtered and dried in a vacuum oven (Isotemp vacuum oven, model 280A, Fisher Scientific) up to 48 h to reach constant weight.

### 2.4. Characterization of nfPLGA and DXM–nfPLGA Composites

The *nf*PLGA and DXM–*nf*PLGA were characterized using several analytical techniques. The synthesized particles hydrodynamic properties were analyzed through Dynamic Light Scattering or DLS (Malvern Nano ZS 90, Model: ZEN 3690, Worcestershire, UK). The functionalized poly(lactic-co-glycolic acid) molecular properties such as weight average molecular weight (M_w_) and number average molecular weight (M_n_) were identified by using Gel Permeation Chromatography or GPC with the Waters Breeze GPC System w/Autosampler at Rutgers Newark NJ. Scanning electron microscopy (SEM) using a JEOL JSM 7900F microscope (JEOL, Tokyo, Japan) was used to image the crystals after carbon coating with an EMS Quorum instrument. The SEM was operated at 1.0 kV at a working distance of 10.0 mm. Additionally, surface elemental composition of *nf*PLGA particles were determined by the Energy Dispersive Spectroscopy (EDS) connected to a SEM instrument. Thermogravimetric analysis (TGA) (PerkinElmer 8000) was used to study nfPLGA incorporation by heating the samples from 30 to 700 °C under a 20 mL/min nitrogen flow at 10 °C/minute. Differential Scanning calorimetry (PerkinElmer DSC 6000) measurements were used to determine the melting point. Raman spectroscopy and microscopy (DXRxi Raman Microscope, Thermo Fisher Scientific, USA) were carried out using a 532 nm laser and gratings and filters. X-Ray diffraction (PANalytical EMPYREAN XRD, Malvern, UK) was performed using a Cu Kα radiation source where determined the crystal structure intensity for 5–70° 2-theta ranges. Additionally, transmission mode Fourier transform infrared (FTIR) spectroscopy analysis was carried out using IR Affinity-1, Agilent Cary 670 Benchtop FTIR instrument (Agilent Technologies, Santa Clara, CA, USA).

The contact angle measurements were performed by recording an image of a water droplet (which acted as the probe) on the solid polymer particles. These images were then analyzed using an online protractor to determine the contact angle [62]. In addition, octanol–water partitioning was studied by placing a drug sample onto a 1:1 ratio of water (aqueous phase) and octanol (organic phase). The mixture was stirred for an hour and allowed to reach equilibrium. The partitioned samples were extracted from the water and octanol phases by ultracentrifugation, and the concentration in each phase was determined using a UV-vis spectrometer to compute the partition coefficient or logP [63].

Dissolution measurements were made using United States Pharmacopeia or USP-42 paddle-II method. A Symphony 7100 Distek instrument (North Brunswick, NJ 08902, USA) was used for this. The pH was set at 1.4 to simulate stomach conditions. The DXM–*nf*PLGA samples were added to a dissolution bath containing 900.0 mL 0.1 N HCl to simulate the pH and dissolution was carried out at 37.5 ± 0.5 °C, rotation speed was set at 75.0 rpm. About 2.0 mL of dissoluted aliquots were transferred from the dissolution medium using needle syringe at 1, 20, 30, 50, 80, 120, 150, 180, and 240 min intervals, then filtered using 0.2 µm syringe filter and analyzed concentration by ultraviolet-visible (UV-vis) measurements. Agilent 8453 (Santa Clara, CA, 95051, USA) model UV-vis spectrophotometer was used for measuring dexamethasone (DXM) absorption at a wavelength at 243.0 nm. Finally, the saturation solubility of the synthesized DXM–nfPLGA composites were determined by stirring the sample in water for 48 h at a room temperature (25 °C) and at pH 7.0.

## 3. Results and Discussion

### 3.1. Synthesis of nfPLGA

The microwave functionalization process altered the PLGA particle properties quite dramatically. This is evident from the photographs in Figure 1. Table 1 presents some of the physicochemical properties of the synthesized *nf*PLGA as compared to the original PLGA. The experimental study found the microwave acid functionalization led to nanosizing and extensive oxidation on the polymer surface. The SEM-EDS analysis showed that the oxygen content increased from 24.76% to 53.07%, implying extensive surface oxidation. Some of the partial ester linkages were broken, which led to more carboxylation and hydroxylation in the synthesized product.

Based on Figure 3b and the physical properties presented in Table 1, the *nf*PLGA nanoparticles were relatively water-dispersible (as high as 4.0 mg/mL) whereas the pure PLGA micron crystals were highly hydrophobic and non-dispersible. The water contact angle (°) was measured by placing a drop of water onto a pile of particles which showed that pure PLGA had a contact angle of 82°, while the *nf*PLGA had a low contact angle of 36°. This clearly demonstrated that the *nf*PLGA was significantly more hydrophilic in nature. The differential scanning calorimetry (DSC) analysis showed that *nf*PLGA had a melting point of 331.78 ℃ and glass transition (T_g_) of 46.14 ℃ which were slightly lower than the original PLGA, which implied that crystallinity was unaltered. Furthermore, gel permeation chromatography (or GPC) analysis found that *nf*PLGA had a lower weight average molecular weight of Mw = 38.3 kDa and number average molecular weight of Mn = 18.4 (a.u). Based on the dynamic light scattering (DLS) analysis, the hydrodynamic diameter of *nf*PLGA particles in water was between 100 and 200 nm with an average (mean) of 161.0 nm with a polydispersity index (PDI) of 0.185. The dispersibility and size distribution of the *nf*PLGA were suitable for *nf*PLGA-drug composite formation.

The powder x-ray diffraction (XRD) analysis of PLGA and *nf*PLGA presented in Figure 2a show similar crystalline peak intensity and demonstrate that crystallinity did not change during microwave functionalization. The RAMAN data presented in Figure 2b (and inset scanned from 800 to 1800 cm^−1^) highlights the functional groups of the PLGA and nfPLGA particles, with bond-stretching for the C-O-C units at 871 cm^−1^, O-C at 1130 cm^−1^, O-C=O at 1454 cm^−1^, C=O at 1766 cm^−1^ and CH2/CH-CH3 at 2948/3000 cm^−1^. It shows a sharp increase in the intensity of the hydroxyl and carbonyl peaks due to increased oxygenation during the microwave oxidation. Moreover, FTIR analysis (Figure 2c) showed an increase in carbonyl peak intensity at 1740 cm^−1^ suggesting increased carboxylic acid functionality. An increase in OH band in the region of 3400–3600 cm^−1^ was also observed.

### 3.2. Characteristics of DXM–nfPLGA Composites 

The scanning electron microscopy (SEM) image of pure DXM and DXM–*nf*PLGA composites are presented in Figure 3. Additionally, the SEM images of PLGA and *nf*PLGA are shown in Figure 3a,b.These images show that the crystal structure of the drug remained unchanged and the *nf*PLGA was successfully incorporated. Figure 3d,e show the presence of *nf*PLGA in a uniform distribution on the surface of the drug crystal, and these were expected to provide the hydrophilic linkages to the aqueous medium, leading to higher dispersibility and solubility.

**Figure 3 nanomaterials-13-00943-f003:**
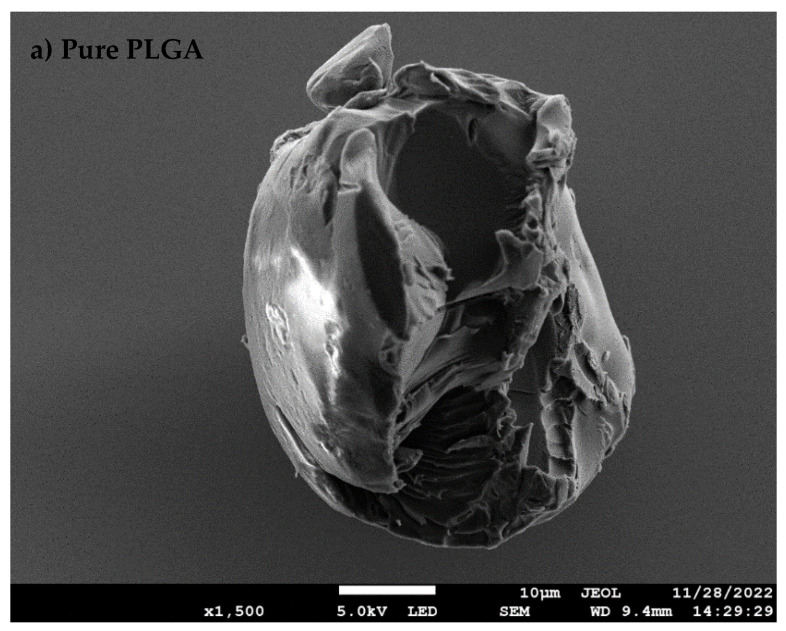
SEM of (**a**) pure PLGA, (**b**) *nf*PLGA, (**c**) pure dexamethasone, (**d**,**e**) DXM–nfPLGA composite.

Figure 4 presents the solubility (mmol/L) and octanol–water partition coefficients of DXM–nfPLGA. The saturation solubility of the formulated drug composites at pH 7.0 showed that that it changed from 0.13 mmol/L for the original drug to 1.89 mmol/L for DXM–nfPLGA. At the same time, the zeta potential, which is used to define colloidal stability, changed from −34.8 for the original drug to −44.3 mV 27 for the DXM–*nf*PLGA. Moreover, the physiological stability of DXM–nfPLGA-1.50 composite was assessed by dispersing particles in different buffer solutions at pH values of 4.0, 6.0, 7.0, and 10.0. No significant changes were observed in the average particle size or mean hydrodynamic diameter at different time intervals of 0, 1, 2, 4, 6, 24 h. This is shown in Figure 4b. This indicated high stability at all pHs. Octanol–water partitioning showed a similar effect, as logP reduced from 1.96 for pure DXM to 0.24 for DXM–*nf*PLGA (Figure 4a).

An important consideration was whether the dexamethasone (DXM) was altered during the composite formation. Figure 5a presents the Raman spectrum of DXM and DXM–*nf*PLGA composites after *nf*PLGA incorporation where the major peak intensities observed for pure dexamethasone were at 688 cm^−1^, 1448 cm^−1^, 1602 cm^−1^, 1658 cm^−1^, 1704 cm^−1^, 2908 cm^−1^, and 2939 cm^−1^. The spectral intensity for the DXM–*nf*PLGA composites shows no variation in these peaks associated with the different functional groups. X-ray diffraction (XRD) analysis (Figure 5b) showed similar crystal structures for DXM and the DXM–*nf*PLGA composites. This was based on the major intensity peak observed at two thetas (2θ) = 6.6°, 7.5°, 9.4°, 10.8°, 12.6°, 13.8°, 14.3°, 15.2°, 15.7°, 17°, 18.6° and so on. Therefore, it is concluded that there was no variation in polymorph. As a result, DXM in the DXM–*nf*PLGA composites are expected to remain biologically similar with increased solubility through the incorporation of inactive *nf*PLGA particles. Additionally, FTIR analysis presented in Figure 6a for the drug composites and pure DXM showed that they were chemically similar. Furthermore, DSC analysis presented in Figure 6b showed similar melting point for both the DXM and nfPLGA incorporated DXM composites.

Additionally, RAMAN mapping and imaging in Figure 7a,b was carried out to see the distribution of the *nf*PLGA on the single drug crustal surface. The distribution map was carried out with a strong peak at 1660 cm^−1^, which corresponded to the carbonyl (C=O) band. The green spots in the images were attributed to the drug crystal surface, while the red was for the *nf*PLGA surface interaction and blue indicated the microscopic image background surfaces. The image showed a non-uniform distribution of *nf*PLGA on the drug crystal surface.

Thermogravimetric analysis (TGA) of *nf*PLGA composites as well as DXM are presented in Figure 8a. These show that the pure drug as well as the DXM–*nf*PLGA composites has similar decomposition profile. The concentrations of *nf*PLGA were also determined from the TGA data. The level of *nf*PLGA incorporation in the different composites were between 0.55% to 1.25%. The melting point (m.p.) data of the different composites are also presented in Table 2 found that the melting point of the original DXM was between 260 and 262 °C, and the DXM–*nf*PLGA composites showed similar values. This also confirms that the polymorph was not altered by *nf*PLGA incorporation, and the composites are thermally stable.

### 3.3. Dissolution of DXM and DXM–nfPLGA

The in vitro dissolution experiment was based on the United States Pharmacopeia or USP-42 dissolution protocol, and the study was conducted in media that mimicked the gastric pH of 1.4. The enhanced dissolution of the drug composites was attributed to the presence of the hydrophilic *nf* PLGA, which led to hydrogen bonding (H-bond) with the API crystal, and eventually enhanced dissolution.

Figure 9 shows the dissolution profile for dexamethasone (DXM) and the *nf*PLGA-incorporated DXM composites. It is clear that *nf*PLGA led to enhanced dissolution rate and aqueous solubility. Table 2 presents the enhanced dissolution rates, as well as the time required to reach 50% (T_50_) and 80% (T_80_) dissolution. Pure DXM showed low solubility, and 100 % dissolution was not possible; however, that was made possible by the incorporation of *nf*PLGA. With the incorporation of 0.75% and 1.5% of *nf*PLGA, T_50_ decreased from 24.0 to 18.0 min and T_80_ reduced from 50.0 min to 35.0 min. Similarly, the initial dissolution rate (or DR) with *nf*PLGA incorporation increased from 289.7 µg/min for pure dexamethasone (DXM) to 513.02 µg/min when the *nf*PLGA incorporation was 1.5%.

## 4. Conclusions

The incorporation of nano-formulated hydrophilic functionalized poly(lactic-co-glycolic acid) or *nf*PLGA significantly enhanced the solubility and the dissolution rate of the hydrophobic drug DXM. The SEM images clearly show the presence of *nf*PLGA dispersed over the surface of the DXM drug crystal. Raman, FTIR, DSC and XRD data point to the fact that the presence of *nf*PLGA did not alter the polymorph and even the melting point remained unaltered. Increase in dissolution rate in the presence of a small amount of the hydrophilic *nf*PLGA was quite pronounced, and consequently the T_50_ and T_80_ values were significantly lower. Finally, the synthesized drug composite particles showed excellent physiological stability at different pH. Mechanistically speaking, we believe that hydrophilic channels produced by *nf*PLGA incorporation enhanced intermolecular interaction with water molecules, and this led to faster dissolution of the API. The use of *nf*PLGA provides an efficient route to drug delivery by increasing the aqueous solubility of hydrophobic molecules. It also requires a minimal amount of the biodegradable polymer, and the process can be easily scaled up. The approach is applicable to other BCS-II and BCS-IV hydrophobic compounds as well. The methodology and the enhanced dissolution are very promising for the drug delivery, and is applicable to bioavailability improvement. Future studies including in vivo measurements are expected to further demonstrate improvement in efficacy.

## Figures and Tables

**Figure 1 nanomaterials-13-00943-f001:**
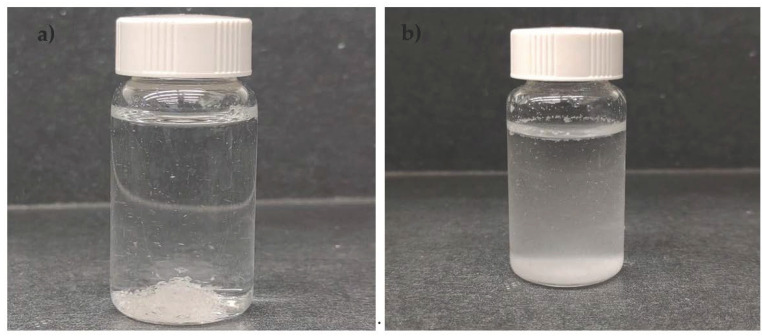
Photograph of (**a**) pure PLGA and (**b**) *nf*PLGA dispersion in water.

**Figure 2 nanomaterials-13-00943-f002:**
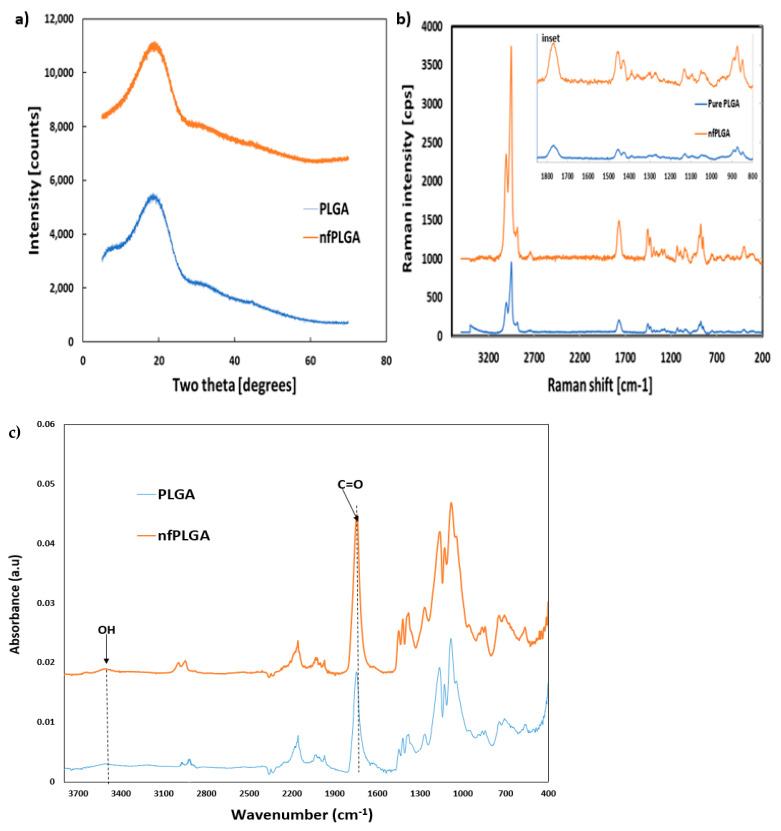
(**a**) XRD, (**b**) RAMAN, and (**c**) FTIR analysis for pure PLGA and *nf*PLGA particles.

**Figure 4 nanomaterials-13-00943-f004:**
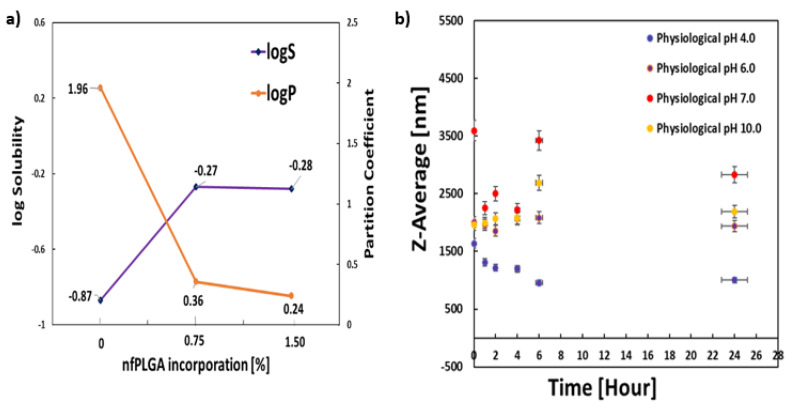
(**a**) Experimental study of solubility and partitioning coefficient and (**b**) Stability of the DMX-*nf*PLGA in a physiological pH buffer.

**Figure 5 nanomaterials-13-00943-f005:**
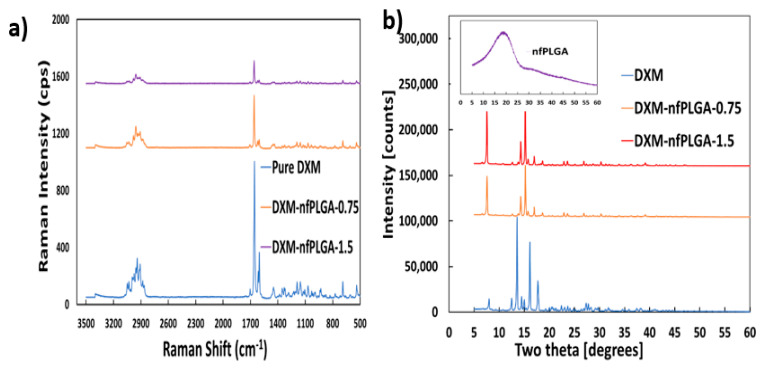
(**a**) DXM–*nf*PLGA composites RAMAN and (**b**) DXM–*nf*PLGA composites XRD analysis.

**Figure 6 nanomaterials-13-00943-f006:**
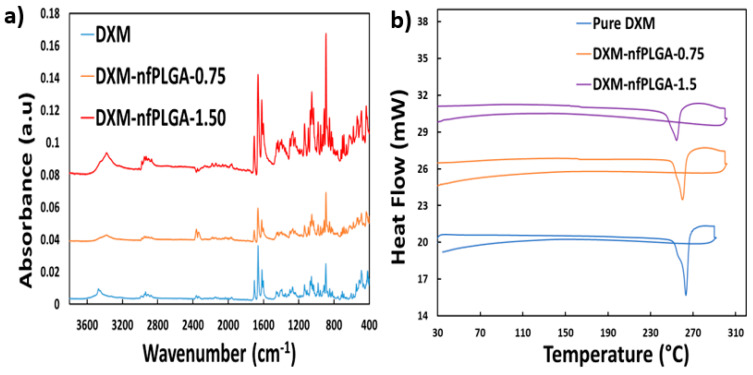
(**a**) DXM–*nf*PLGA composites FTIR and (**b**) DXM–*nf*PLGA composites DSC analysis.

**Figure 7 nanomaterials-13-00943-f007:**
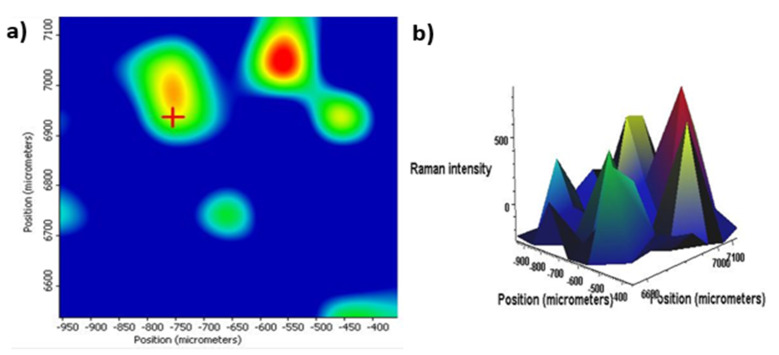
(**a**) Raman mapping of drug crystal surface and (**b**) 3D images of mapped DXM–*nf*PLGA Composites.

**Figure 8 nanomaterials-13-00943-f008:**
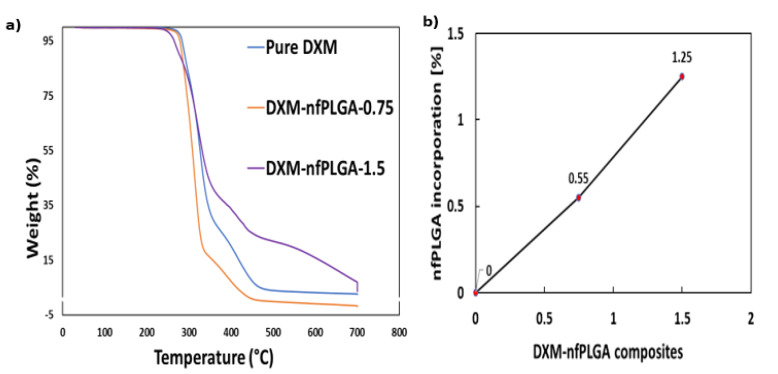
(**a**) DXM–*nf*PLGA composites TGA analysis and (**b**) % *nf*PLGA incorporation.

**Figure 9 nanomaterials-13-00943-f009:**
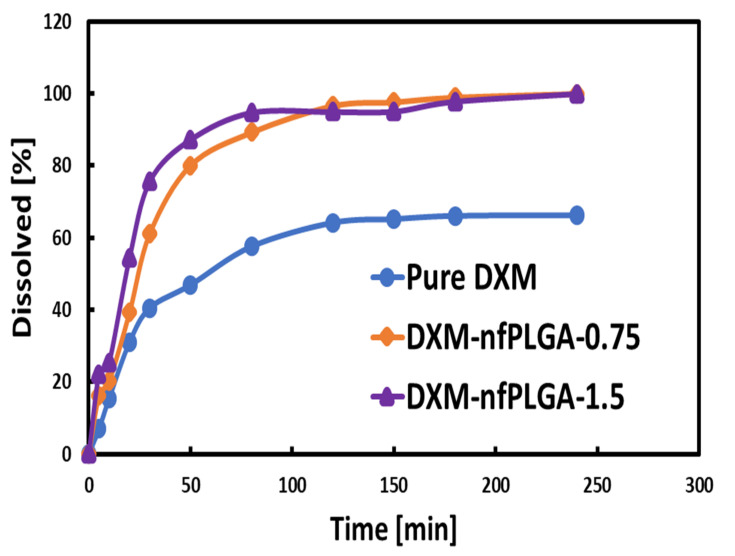
Dissolution profile for DXM–nfPLGA composites.

**Table 1 nanomaterials-13-00943-t001:** Physical properties of PLGA and *nf*PLGA particles.

Properties	Original PLGA	*nf*PLGA
Carbon (wt. %)	75.24	46.93
Oxygen (wt. %)	24.76	53.07
M_w_	47.9	38.3
M_n_	34.1	18.4
Particle size (nm)	large	~161.0 nm
Zeta potential [mV]	−13.1	−31.7
T_m_ [°C]	338.03	331.78
T_g_ [℃]	49.01	46.14
Dispersibility [mg/mL]	Non-disperse	4
Contact angle [°]	82	36

**Table 2 nanomaterials-13-00943-t002:** Physicochemical properties and dissolution data related to DXM–*nf*PLGA composite.

Formulations	(T_50_)	(T_80_)	DR	MP	Z.P
	(min)	(min)	(µg/min)	(°C)	[mV]
Pure DXM	57	n.d.	289.7	261.74	−17.2
DXM–*nf*PLGA-0.75	24	50	422.4	260.22	−34.8
DXM–*nf*PLGA-1.5	18	35	513.02	259.38	−44.3

[Abbreviation: n.d. = not dissolved; *nf* = nano functional; DR = Dissolution rate; MP = Melting point; logP = logarithm of partition].

## Data Availability

Research data will be available from the authors upon request.

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
