# Peer review of "Synthesis of Microwave Functionalized, Nanostructured Polylactic Co-Glycolic Acid (*nf*PLGA) for Incorporation into Hydrophobic Dexamethasone to Enhance Dissolution"

_nanomaterials, 2023, doi:10.3390/nano13050943_

Round 1

Reviewer 1 Report

In this manuscript, the authors initially functionalized hydrophobic PLGA into the partially hydrophilic nanostructured PLGA (nfPLGA) by a microwave-assisted method, then nfPLGA was incorporated into the dexamethasone which as the result, improving its solubility in water.

The present work is generally a good example for reference for other researchers and the manuscript was also well prepared. However, some small typos (for example, "The SEM images of PLGA and nfPLGA are shown in Figure 1a and b") still need to be revised accordingly; The figures and tables could be improved more to meet the same style. In general, I recommend accepting this manuscript after minor revision.

Author Response

Comments and Suggestions for Authors

In this manuscript, the authors initially functionalized hydrophobic PLGA into the partially hydrophilic nanostructured PLGA (nfPLGA) by a microwave-assisted method, then nfPLGA was incorporated into the dexamethasone which as the result, improving its solubility in water.

The present work is generally a good example for reference for other researchers and the manuscript was also well prepared. However, some small typos (for example, "The SEM images of PLGA and nfPLGA are shown in Figure 1a and b") still need to be revised; accordingly, The figures and tables could be improved more to meet the same style. In general, I recommend accepting this manuscript after minor revision.

Response: Thanks for suggestion. We have revised and corrected in line 203. All the figures and Tables are now improved to same style.

Reviewer 2 Report

Dear authors,

The manuscript describes a nice way to produced nanostructured functionalized PLGA alginate. Moreover, the authors investigated its application in increasing the solubility of dexamethasone by an anti-solvent precipitation. I think that the authors have provided a really thorough characterization. This information is valuable to develop further applications of this type of composite. I recommend the publication in Nanomaterials, after minor revision.

My comments concerning this manuscript are following:

-      The proposed functionalization/degradation of PLGA scheme should be added. Furthermore, the sample codes in the figures and table should be maintained as defined in the text (DXM-nfPLGA).

-       In section 2.4, “polylactic co glycolic acid” must be written correctly such as “poly(lactic-co-glycolic acid)”.

-  Explanation of RAMAN result is weak. Please add the detailed information of RAMAN results. Please insert, also, the information of each peak in RAMAN pictures.

-       Conclusion should be more explicit.

Author Response

Comments and Suggestions for Authors

Dear authors,

The manuscript describes a nice way to produced nanostructured functionalized PLGA alginate. Moreover, the authors investigated its application in increasing the solubility of dexamethasone by an anti-solvent precipitation. I think that the authors have provided a really thorough characterization. This information is valuable to develop further applications of this type of composite. I recommend the publication in Nanomaterials, after minor revision.

My comments concerning this manuscript are following:

-      The proposed functionalization/degradation of PLGA scheme should be added. Furthermore, the sample codes in the figures and table should be maintained as defined in the text (DXM-nfPLGA).

Response: This has been added in the revised manuscript line 54-56, 93-94.

-       In section 2.4, “polylactic co glycolic acid” must be written correctly such as “poly(lactic-co-glycolic acid)”.

Response: Correction made as suggested.

-  Explanation of RAMAN result is weak. Please add the detailed information of RAMAN results. Please insert, also, the information of each peak in RAMAN pictures.

Response: Updated in revised draft line 230-233, 263-266.

-       Conclusion should be more explicit.

Response: The conclusions section has been updated.

Reviewer 3 Report

The manuscript presents a novel and interesting solubility enhancement method using nanostructured, functionalized PLGA (nfPLGA). However, to support the publication of this work as a "communication," we suggest the following revisions:

 1. Section 2.4 should be expanded to include more detailed information about the determination of solubility and partition coefficient, as well as contact angle measurements.

 2. While the experimental procedure was outlined, DSC and IR results were not presented. We suggest adding these data as they are important to support the authors' conclusions.

 3. Table 1 shows that the molecular weight of nfPLGA is smaller than PLGA, and the authors attribute this to oxidation. However, it appears to be degradation of PLGA. Additionally, the dispersibility results do not match the results and discussion. We suggest revising the text to clarify these points.

 4. In Figure 2b, we suggest normalizing the Raman spectra to highlight any significant changes between PLGA and nfPLGA.

 5. Figure 5b shows that the XRD diffractogram for DXM-nfPLGA is different from DXM alone, but the authors state that there is no significant change in crystal structure. To support their conclusion, we suggest including pXRD diffractograms of nfPLGA alone and nfPLGA and DXM mixture.

We hope these suggestions will improve the manuscript and look forward to seeing the revised version.

Author Response

Comments and Suggestions for Authors

The manuscript presents a novel and interesting solubility enhancement method using nanostructured, functionalized PLGA (nfPLGA). However, to support the publication of this work as a "communication," we suggest the following revisions:

  1. Section 2.4 should be expanded to include more detailed information about the determination of solubility and partition coefficient, as well as contact angle measurements.

Response: This has been added in the manuscript (line 175-182, 193-195).

  1. While the experimental procedure was outlined, DSC and IR results were not presented. We suggest adding these data as they are important to support the authors' conclusions.

Response:  This has been added as the new figure 6a and 6b and described in line 273-274, 277-278.

  1. Table 1 shows that the molecular weight of nfPLGA is smaller than PLGA, and the authors attribute this to oxidation. However, it appears to be degradation of PLGA. Additionally, the dispersibility results do not match the results and discussion. We suggest revising the text to clarify these points.

Response: The explanation has been revised (line 206-207).  

  1. In Figure 2b, we suggest normalizing the Raman spectra to highlight any significant changes between PLGA and nfPLGA.

Response: Please see lines 229-232 and updated figure 2b with inset scanned data.

  1. Figure 5b shows that the XRD diffractogram for DXM-nfPLGA is different from DXM alone, but the authors state that there is no significant change in crystal structure. To support their conclusion, we suggest including pXRD diffractograms of nfPLGA alone and nfPLGA and DXM mixture.

Response: This has been added in the revised draft figure 5b and line 267-270.

We hope these suggestions will improve the manuscript and look forward to seeing the revised version.

Response: Thanks for the suggestion, we have revised accordingly. 

Reviewer 4 Report

The poor solubility of pharmaceutical drugs still seriously limits their clinical development. Various approaches have been proposed including microencapsulation and lipid-based technologies to improve drug solubility.
The present study deals with the synthesis of surface-functionalized polylactic co-glycolic acid (PLGA) nanoparticles for incorporation into corticosteroid dexamethasone (DMX) in orde
r to improve in vitro dissolution profile. PLGA crystals were submitted to a strong acid and microwave treatment that led to a high degree of oxidation. The resulting PLGA (nfPLGA) was more water dispersible compared to the original PLGA.

Specific comments.

- A major concern is the lack of information regarding the molecular mechanism responsible for the DMX improved solubility. The formation of hydrophilic channels produced by nfPLGA incorporation is entirely speculative and not supported by any convincing experimental argument. Which interactions are responsible for the stability of the DMX-nfPLGA complex is another key point to be addressed.

- Figure 5a- Raman spectrum of DXM and DXM-nfPLGA reveals that DXM was not structurally altered during complex formation. However, is DXM still biologically active as a  DXM-nfPLGA complex?

- Improvement of DMX solubility is quite modest. Is 1,5,% a limit for nfPLGA incorporation.

-  What is the stability of the DMX-nfPLGA complex in a physiological environment..

- LIposomal dexamethasone targeted several primary cell types in a dose and time-dependent manner and specifically reduced cytotoxicity against human fibroblasts and macrophages in comparison to the solute drug. Discussing the advantages of using DMX-nf PLGA compared with liposomal dexamethasone will be quite informative and contribute to improving the impact of the manuscript.

Author Response

Comments and Suggestions for Authors

The poor solubility of pharmaceutical drugs still seriously limits their clinical development. Various approaches have been proposed including microencapsulation and lipid-based technologies to improve drug solubility.
The present study deals with the synthesis of surface-functionalized polylactic co-glycolic acid (PLGA) nanoparticles for incorporation into corticosteroid dexamethasone (DMX) in order to improve in vitro dissolution profile. PLGA crystals were submitted to a strong acid and microwave treatment that led to a high degree of oxidation. The resulting PLGA (nfPLGA) was more water dispersible compared to the original PLGA.

Specific comments.

- A major concern is the lack of information regarding the molecular mechanism responsible for the DMX improved solubility. The formation of hydrophilic channels produced by nfPLGA incorporation is entirely speculative and not supported by any convincing experimental argument. Which interactions are responsible for the stability of the DMX-nfPLGA complex is another key point to be addressed.

Response: Thanks for this question. Please see the mechanism depicted in graphical abstract. We have discussed the stability in the revised daft in line 251-257.

- Figure 5a- Raman spectrum of DXM and DXM-nfPLGA reveals that DXM was not structurally altered during complex formation. However, is DXM still biologically active as a DXM-nfPLGA complex?

Response: DXM in the composites will remain biologically active and this has been mentioned in line 270-273.

- Improvement of DMX solubility is quite modest. Is 1,5,% a limit for nfPLGA incorporation.

Response: Based on previous work and precipitation experiments, 1-2% of nfPLGA incorporation was accomplished and used in this study. Future effort may include higher concentrations of the nfPLGA.

-  What is the stability of the DMX-nfPLGA complex in a physiological environment.

Response: This is presented in lines 252-256 and in figure 4b.

- Liposomal dexamethasone targeted several primary cell types in a dose and time-dependent manner and specifically reduced cytotoxicity against human fibroblasts and macrophages in comparison to the solute drug. Discussing the advantages of using DMX-nf PLGA compared with liposomal dexamethasone will be quite informative and contribute to improving the impact of the manuscript.

Response: Additional details have been presented in Lines 100-104.

Round 2

Reviewer 3 Report

Overall, I believe that the manuscript requires major revisions before it can be considered for publication. Below are some of my specific concerns:

 1. The authors claim that there is no significant change in crystal structure in DXM-nfPLGA compared to DXM alone, but Figure 5b shows that the XRD diffractogram for DXM-nfPLGA is completely different from that of DXM alone. In particular, there is no major peak at 2 theta = 10. Therefore, the authors should provide a more accurate and detailed description of the crystal structure of DXM-nfPLGA.

 2. In line 113, the authors state that DXM and PLGA form a co-crystal. However, based on Figure 3 and Figure 7, it appears that PLGA is simply adsorbed to DXM crystal rather than co-crystallized with it. Therefore, the authors should clarify their terminology and provide a more accurate description of the interaction between DXM and PLGA.

 3. In Figure 6b, the thermogram of pure DXM shows a shoulder that is different from the other samples. The authors should explain the significance of this observation and how it relates to their overall findings.

 4. The authors suggest that the increased COOH affects the solubility of DXM, but they have not checked the band ratio of the hydroxyl group and carbonyl group of carboxylic acid in FT-IR. This is an important analysis that should be included in the manuscript.

 5. The authors use UV spectrometry to measure solubility and dissolution, but they have not taken into account the fact that PLGA may have UV absorption properties similar to PLA (https://www.mdpi.com/1996-1944/12/3/481). Therefore, it is important to confirm the dissolution enhancement using HPLC, which is introduced in USP. The authors should perform these additional experiments and present their results in the revised manuscript.

 I hope these comments are helpful in improving the quality of the manuscript, and I look forward to seeing the revised version.

Author Response

Review Report (Reviewer 3)

Comments and Suggestions for Authors

Overall, I believe that the manuscript requires major revisions before it can be considered for publication. Below are some of my specific concerns:

  1. The authors claim that there is no significant change in crystal structure in DXM-nfPLGA compared to DXM alone, but Figure 5b shows that the XRD diffractogram for DXM-nfPLGA is completely different from that of DXM alone. In particular, there is no major peak at 2 theta = 10. Therefore, the authors should provide a more accurate and detailed description of the crystal structure of DXM-nfPLGA.

Response: Thank you for catching this. We have repeated XRD and increased the scanning steps number and scan time to correctly to measure the low intensity 2 theta range peaks and presented correct theta value in the revised draft.

  1. In line 113, the authors state that DXM and PLGA form a co-crystal. However, based on Figure 3 and Figure 7, it appears that PLGA is simply adsorbed to DXM crystal rather than co-crystallized with it. Therefore, the authors should clarify their terminology and provide a more accurate description of the interaction between DXM and PLGA.

Response: We agree they are not co-crystals; the correction has been made as suggested. Thanks for this suggestion.

  1. In Figure 6b, the thermogram of pure DXM shows a shoulder that is different from the other samples. The authors should explain the significance of this observation and how it relates to their overall findings.

Response: In DSC analysis, this shoulder for melting is minor and this could be possible due to the ±2/3 % impurity present in supplied drug samples. However, the antisolvent precipitated DXM-nfPLGA composite found highly crystalline pure materials and no shoulder was observed. However, DSC analysis is significant for this study because it relates to the melting point, purity, polymorphic behavior and thermal stability.

  1. The authors suggest that the increased COOH affects the solubility of DXM, but they have not checked the band ratio of the hydroxyl group and carbonyl group of carboxylic acid in FT-IR. This is an important analysis that should be included in the manuscript.

Response: FTIR analysis has been added for PLGA in the revised draft figure 2c and line 234-236.

  1. The authors use UV spectrometry to measure solubility and dissolution, but they have not taken into account the fact that PLGA may have UV absorption properties similar to PLA (https://www.mdpi.com/1996-1944/12/3/481). Therefore, it is important to confirm the dissolution enhancement using HPLC, which is introduced in USP. The authors should perform these additional experiments and present their results in the revised manuscript.

Response: We have checked the UV profile as well as potential photolytic degradation of nfPLGA. No noticeable absorption was observed for the nfPLGA at the 243 nm wavelength where the measurements were made. Therefore, we do not believe HPLC measurements were necessary here.